# Device-Aided Therapies in Parkinson’s Disease—Results from the German Care4PD Study

**DOI:** 10.3390/brainsci13050736

**Published:** 2023-04-28

**Authors:** Odette Fründt, Anne-Marie Hanff, Annika Möhl, Tobias Mai, Christiane Kirchner, Ali Amouzandeh, Carsten Buhmann, Rejko Krüger, Martin Südmeyer

**Affiliations:** 1Department of Neurology, Klinikum Ernst von Bergmann, Charlottenstraße 72, 14467 Potsdam, Germany; 2Transversal Translational Medicine, Luxembourg Institute of Health (LIH), 1A-B, Rue Thomas Edison, L-1445 Strassen, Luxembourg; 3Faculty of Science, Technology and Medicine, University of Luxembourg, Belval Campus, Maison du Savoir, 2 Avenue de l’Université, L-4365 Esch-sur-Alzette, Luxembourg; 4Department of Epidemiology, CAPHRI School for Public Health and Primary Care, Maastricht University Medical Centre, Postbus 5800, 6202 AZ Maastricht, The Netherlands; 5Institute of Medical Biometry and Epidemiology, University Medical Center Hamburg-Eppendorf, Martinistraße 52, 20246 Hamburg, Germany; 6Department of Nursing Development/Nursing Research, University Hospital Frankfurt, Theodor-Stern-Kai 7, 60590 Frankfurt, Germany; 7Department of Neurology, University Medical Center Hamburg-Eppendorf, Martinistraße 52, 20246 Hamburg, Germany; 8Luxembourg Centre for Systems Biomedicine (LCSB), University of Luxembourg, 6, Avenue du Swing, L-4367 Belvaux, Luxembourg; 9Parkinson Research Clinic, Centre Hospitalier de Luxembourg (CHL), 4 Rue Nicolas Ernest Barblé, L-1210 Luxembourg, Luxembourg; 10Department of Neurology, University Medical Center Düsseldorf, Moorenstraße 5, 40225 Düsseldorf, Germany

**Keywords:** Parkinson’s disease, device-aided therapies, deep brain stimulation, medication pump, infusion therapies, advanced, disease progression

## Abstract

Data on the use of device-aided therapies (DATs) in people with Parkinson’s disease (PwP) are scarce. Analyzing data from the Care4PD patient survey, we (1) evaluated application frequency and type of DAT in a larger, nationwide, cross-sectoral PwP sample in Germany; (2) analyzed the frequency of symptoms indicative for advanced PD (aPD) and need for DAT amongst the remaining patients and (3) compared the most bothersome symptoms and need for professional long-term care (LTC) of patients with and without suspected aPD. Data from 1269 PwP were analyzed. In total, 153 PwP (12%) received DAT, mainly deep brain stimulation (DBS). Of the remaining 1116 PwP without DAT, >50% fulfilled at least one aPD criterion. Akinesia/rigidity and autonomic problems were most bothersome for PwP with and without suspected aPD, with more tremor in the non-aPD and more motor fluctuations and falls in the aPD group. To recapitulate, the German DAT application rate is rather low, although a large proportion of PwP fulfills aPD criteria indicating a need for intensified treatment strategies. Many reported bothersome symptoms could be overcome with DAT with benefits even for LTC patients. Thus, precise and early identification of aPD symptoms (and therapy-resistant tremor) should be implemented in future DAT preselection tools and educational trainings.

## 1. Introduction

People with advanced Parkinson’s disease (aPD) are characterized by motor fluctuations such as off-phases or dyskinesia as well as non-motor complications, noticeably reducing their quality of life (QoL, [1]). If oral medication is not sufficient for controlling these symptoms, more advanced therapeutic strategies are required. In 2018, a Delphi panel consensus by Antonini and colleagues recommended the 5-2-1 criteria as a possible standardized classification scheme to screen for and diagnose aPD [2,3]: (a) 5 or more daily dosages of oral levodopa; (b) 2 or more hours of clinical “off” time per day; (c) 1 or more hour(s) per day with troublesome dyskinesia. 

These criteria show moderate consensus with physician judgements [1] and are useful for identification of patients with the need for treatment optimization [3]. This includes the use of device-aided therapies (DATs) [4] such as deep brain stimulation (DBS), or infusion therapies (continuous subcutaneous apomorphine infusion (CSAI) or levodopa carbidopa gastrointestinal gel (LCIG)) that show positive effects on the patients’ QoL even in long-term observations [5,6,7,8,9,10,11]. 

However, DATs are only rarely applied or considered lately in the course of the disease. In a recent study including 177 German inpatient people with Parkinson’s disease (PwP), about half of them were classified by physicians as having aPD. Of these patients, 58% were identified to be eligible for DAT, whereas only 41% received it [12]. This indicates—as compared to international standards—a tendency in Germany towards more complex oral medication regimens for aPD patients [12], with reluctance regarding the start of DAT. Delayed aPD diagnosis and inappropriately continued oral medication might, however, result in reduced symptom control and QoL in these patients. Some aPD symptoms, such as impaired motor functioning with “off” times, but also non-motor illusions or cognitive impairment (that are also mentioned in the Delphi criteria [2,3]) are even predictors for the need of long-term care (LTC) in PwP [13]. Furthermore, patients with aPD experience a higher risk of nursing home admission than those without, while admissions could be reduced or delayed with DAT [14]. 

Our current questionnaire-based study aimed to, firstly, evaluate the application frequency and type of DAT in a larger, nationwide, cross-sectoral PwP sample in Germany. Secondly, we wanted to analyze the frequency of symptoms that are indicative for aPD and the possible need for DAT amongst the remaining patients under medical treatment, only based on the 5-2-1 criteria. Thirdly, we were interested in the most bothersome symptoms and need for professional LTC of patients with and without suspected aPD to understand possibly different (unmet) needs of both groups.

## 2. Materials and Methods

We analyzed data from our nationwide, cross-sectoral, comprehensive, survey-based Care4PD study [15,16]. This study consists of a patient questionnaire (presented here) and a nursing staff questionnaire (not presented here). The patient questionnaire was previously introduced in more detail elsewhere [17] and the data collection was realized from March to July 2021. Study participation was voluntary and anonymous.

The Care4PD patient questionnaire was originally developed with the aim to evaluate the overall care situation of a larger group of PwP in Germany across all sectors. It consisted of 56 questions in total and can be found as the complete version (English translation) in the Appendix A of this manuscript.

Following a short introduction and explanation of the questionnaire including instructions on how to fill it out properly (page 1), the final patient survey was divided into two sections: a general section for all PwP (questions #1–40) and a specific part for only those participants with Parkinson’s disease (PD) receiving professional long-term care (questions #41–56). Regarding the content, besides general demographic data it focused on disease characteristics such as disease duration and severity, the most bothersome symptoms (multiple choice with 3 answers allowed), applied treatments including medication and/or DAT, information regarding medical and therapeutical support, problems with activities of daily living, the need for and type of professional long-term care (LTC; e.g., outpatient care services, professional domestic 24-h care or residents of nursing homes), information on quality, quantity and costs of nursing care, the PwPs’ quality of life and the influence of the COVID-19 pandemic on PwPs’ care situation. 

In general, instructions and explanations were given in written form with simple, comprehensible wording that has been recommended by a specialist experienced in gerontology with a focus on barrier-free questionnaires (Josephine Green [18]). Additionally, we followed the recommendations of the guidelines to optimize questionnaires [19] and consulted a statistician for facilitating optimal statistical analysis of questions. With respect to questions belonging to characteristics of the Parkinson’s disease (PD), e.g., PD symptoms or quality of life, questionnaire development was guided by internationally validated questionnaires or scales. These were also integrated—at least partially—into our survey, e.g., the Hoehn and Yahr scale (question #7, [20]) that has previously been used for self-rating of disease severity [21,22], a single item of the MDS-Unified Parkinson’s Disease Rating Scale (MDS-UPDRS) part IV for motor “off” complications (#9, [23]), the 8-item version of the Parkinson’s Disease Questionnaire (PDQ-8) for quality of life (#10 presented as total score of all items, [24]) or the Katz index for problems with activities of daily living (#13 presented as total score of all items, [25]) and have previously been used manifold internationally. The initial questionnaire was than tested in a pilot phase with 10 PD patients and 5 neurologic PD specialists and was later adapted according to their suggestions, again with the help of the gerontologist mentioned above, to create the final version that all test subjects agreed with.

The questionnaire was mainly addressed to PwP, but their family members, relatives, nursing staff or another third person were requested to support the patient in answering it (e.g., in case of cognitive impairment) and information on the person that predominantly filled out the questionnaire was gathered (question #1).

It was distributed via the members journal of the German PD association and was sent to several in- and outpatient institutions in rural and urban German areas specializing in PD with a total circulation of about 25,000 copies. For this manuscript, we analyzed a subset of items out of the comprehensive 56 questions survey that, content-wise, related to aspects belonging to advanced PD and/or DAT (see blue labeling in Appendix A for concrete content of questions). Single- or multiple-choice questions, graduated (Likert) scales, visual analogue scales (from 0 = not applicable/not at all to 10 = very applicable/very much) and open questions were used (see Appendix A).

Returned questionnaires (*n* = 1437, response rate 5.7%) were scanned, automatically recorded using the FormPro software version 3.0 by OCR Systeme GmbH; Leipzig, Germany [26] with good quality (see [17]) and manually checked for plausibility. Questionnaires that were sent twofold, were inconsistent or had >30% missing data were excluded.

Taking under account the 5-2-1 criteria mentioned above [2,3], we defined the following aPD-suspect criteria: (a)A medication frequency of >6 times per day (considering the intake of additional dopaminergic rescue medication);(b)An “off” time >25% during waking time (about 4 h in total of a 16 h waking time);(c)The existence of troublesome dyskinesia.

Criteria (b) and (c) were both based on the Movement Disorders Society Unified Parkinson’s Disease Rating Scale (MDS-UPDRS) questionnaire [23]). 

Results were described as absolute numbers (n), relative frequencies (% of available data; missing data were excluded from analysis) or means with standard deviation (SD) and ranges from minimum to maximum. Group comparisons were performed between i) patients with and without DAT and ii) those with (“aPD group”; who fulfilled at least one of three aPD criteria) and without (“n-aPD group”, none of the criteria fulfilled) suspected aPD using Student’s *t*-test or Pearson’s Chi squared test. All analyses were performed in IBM SPSS^®^ Statistics version 27, Armonk, North Castle, United States [27] and were explorative.

The study was approved by the local ethics committee of the medical council of Brandenburg (reference number: S10(bB)/2021 with amendment) and is in accordance with the Declaration of Helsinki. 

## 3. Results

A total of 1269 out of 1437 returned questionnaires (88%) were included. Questionnaires with inconsistent answers (*n* = 116) or >30% missing data (*n* = 52) were excluded from analysis. In most cases (1030 out of 1238 patients = 83.2% (*n* = 31 missing or multiple answers)) the patients themselves predominantly answered the questionnaire. In 16.3%, the family members/relatives filled it out, and in only a small proportion of 0.5%, either another third person or the patients’ nursing staff replied (question #1).

### 3.1. Participants with Device-Aided Therapies 

In total, 153 out of 1269 participants (12%, 56% of them were male) received DAT (question #14). This group was younger (68.0 ± SD 8.8 vs. 73.2 ± 8.5 years), had a younger age at diagnosis (51.0 ± SD 9.6 vs. 63.8 ± 9.6 years) and a higher disease severity according to Hoehn and Yahr score (3.4 ± SD 0.9 vs. 2.9 ± 1.1; all *p* < 0.001, question #7) compared to those without DAT. Furthermore, 22% (*n* = 34) of participants with DAT received professional long-term care (question #23). Most of them used DBS (*n* = 121, 79%), followed by LCIG (*n* = 21, 14%) and CSAI (*n* = 7, 5%, question #14). Four participants received a combination of different DATs (three with DBS + CSAI and one with DBS + LCIG). The subjects’ own or a family members’ capacities to handle the DATs were evaluated to be relatively good, with a mean of 7.5 on a scale from 0 (not at all) to 10 (very good, question #19).

### 3.2. Participants without Device-Aided Therapies

#### 3.2.1. Indications for aPD and the Possible Need for Device-Aided Therapies

We recognized a high number of patients reporting symptoms that are indicative for aPD, with a total of 627 out of 1116 PwP (56%), fulfilling at least 1 of our 3 aPD-suspect criteria:(a)Medication frequency (question #15): Oral medication was used in nearly all PwP (99%, *n* = 1107/1116) with good therapy adherence in 96% of PwP (*n* = 1070/1097), who reported taking their medication “always” or “often” sticking to their medication scheme. A total of 56% of PwP took their oral medication 4 to 6 times per day (*n* = 626/1113) and about 21% (*n* = 234/1113) even had a medication frequency of more than 6 times.(b)Amount of “off” time per day (question #9): Off phases, in general, were recognized by the majority of patients (*n* = 736/1089 = 68%, (remaining *n* = 231 “no off” or *n* = 122 “don’t know”)), of which 454/1089 patients (42%) suffered from >25% “off” time during waking hours, which is about 4 h per day when assuming about 16 h of waking time.(c)Troublesome dyskinesia (question #8, item “involuntary movements”) was reported in 15% of PwP (*n* = 170/1108).

Regarding clinical characteristics, the aPD group (those who fulfilled at least one of the aPD-suspect criteria mentioned in the Section 2) was older, had an earlier disease onset, an increased disease severity (question #7), needed more help with activities of daily living (ADL, indicated by lower Katz index, question #13), was more often served by professional long-term care (LTC, question #23) and had a lower QoL (question #10) compared to those without suspected aPD (Table 1).

#### 3.2.2. Most Bothersome Symptoms of PwP with and without Suspected aPD (Subitems of Question #8)

In total, 906/1116 patients replied to this question correctly (81%) with an allowed maximum of 3 answers (no answer: *n* = 8, >3 answers: *n* = 202). In both groups, akinesia/rigidity was ranked to be the most bothersome symptom (aPD: 53.3%, n-aPD: 41.1%) and autonomic problems were also ranked among both groups’ three most troublesome symptoms (aPD: 34.8%, n-aPD: 32.1%). Furthermore, different symptom constellations became apparent between PwP with and without suspected aPD (see Figure 1). In group comparison, the n-aPD group reported more troublesome tremor (*p* < 0.001) which was the second most bothersome symptom in them (35.5%), whereas those with suspected aPD reported more often about akinesia/rigidity (*p* < 0.001), motor fluctuations (*p* < 0.001) and gait disorder with falls (*p* = 0.007).

## 4. Discussion

Our questionnaire-based Care4PD study evaluated the application frequency, type, and possible indications for DAT as well as most bothersome symptoms in a larger, nationwide, cross-sectoral sample of PwP. Our results indicate three key messages that are discussed in detail below: (1)About 12% of rather younger, more severely affected PD patients with earlier disease onset already receive DAT with preference for DBS.(2)Of the remaining patients, more than 50% show evidence for at least one of three aPD-suspect symptoms, indicating higher numbers of aPD patients as possible candidates for DAT.(3)The most bothersome symptom profiles of PwP with and without suspected aPD as well as their need for professional LTC vary between groups, possibly indicating different diagnostic and care demands.

### 4.1. Participants with Device-Aided Therapies

In total, 12% of PwP reported that they already receive DAT, with DBS representing the device mostly used (79%) followed by LCIG (14%) and CSAI (5%). To our knowledge, these are the first data on nationwide, cross-sectoral, DAT application frequency and type in Germany. The DAT type (DBS 57% > LCIG 39% > CSAI 8% [1]) and application rate are comparable to international data (15.2% [1]) with, however, a slightly lower frequency and more prominent use of DBS in our German cohort. Even in previous studies including PwP who are about to start DAT, the German rate (17.6% [12]) was lower in international comparison (22.6% [1]), although all of these DATs are effective for motor and non-motor symptom reduction [28,29,30]. Apparently, despite increasing caseloads during recent years [31], in Germany, referral to and application of these treatments [12,32] are still reluctantly given, with possible reasons discussed below. 

Here, DATs were mainly applied in younger, more severely affected PwP with a longer disease duration. This may be influenced by the seemingly preferential use of DBS, possibly resulting from the positive findings of the Early Stim study [33] or because of contraindications such as cognitive impairment linked to older age. However, as infusion therapies especially [34] (but still DBS [35]) can be helpful in elderly PwP, we should rethink indications and timing for DAT and should train players involved in decision making to choose a certain DAT more individually dependent on specific patient characteristics [34] and symptom constellations as different devices target different symptoms [34,36]. Interestingly, CSAI was only rarely used, although efficacy of this least invasive, reversable option has been proven manifold (review: [34,36,37]) with even lower lifetime costs compared to other DATs [38]. Possibly, individual contraindications or side effects may play a role [39,40]. The reluctant CSAI application [41] could possibly be overcome with more information and education in the future. Alternatively, newer, less invasive, but potentially more tolerable approaches (at least for higher-aged PwP) such as the Levodopa–Entacapone–Carbidopa intestinal gel infusion (LECIG, [42,43]) or the subcutaneous, continuous Foslevodopa/Foscarbidopa infusion [44,45,46] or the potentially neuromodulation-focused ultrasound [47] might offer novel therapeutic avenues.

Although patients rated their own (or a family members’) DAT handling capacities to be quite good, the fact that 22% of them received LTC indicates the need to also train the LTC nursing staff to handle these devices.

### 4.2. Participants without Device-Aided Therapies

#### 4.2.1. Indications for aPD and the Possible Need for Device-Aided Therapies

More than 50% of the remaining participants without DAT fulfilled at least 1 of 3 aPD-suspect symptoms. This indicates a high prevalence of aPD patients that is comparable to previous studies in PwP presenting to movement disorder specialists in Germany (122/177 patients = 68.9%) and internationally (1342/2615 patients = 51.3%). About 20% of PwP took their medication >6 times per day, and more than 40% of PwP suffered from >25% (4 h) “off” time during waking time, although therapy adherence was reported to be very good and troublesome dyskinesia was reported in about 15% of PwP.

Nevertheless, our results indicate a gap between a high proportion of aPD-suspect patients and those actually receiving DAT. This is in line with previous (inpatient) German findings showing that the number of DAT-eligible patients is higher than that with actual referral to and application of these treatments, especially DBS [12,32], but is now initially visible in our larger nationwide, cross-sectoral patient collective including all types of DATs. Nevertheless, the transition from oral medication to DAT in aPD might be important, as DAT-receiving PwP show less motor and non-motor impairments than DAT-eligible PwP without these treatments [1]. Furthermore, DAT application might reduce or even prevent care dependency [14]. The latter is already given in 27.6% of our aPD-suspect sample, and in 12.7% of n-aPD participants that needed professional long-term care. Higher DAT application rates might therefore even be more economical with decreased lifetime costs [11,38,48,49].

Possible reasons for a more sophisticated “fine tuning” of oral PD medication in German aPD patients [12] compared to international approaches finally leading to delayed or no DAT initiation are various. They include (i) a reduced detection of aPD and those PwP with the need of DAT, possibly resulting from general gaps of knowledge about DATs amongst health care providers such as neurologists and (non-specialist) physicians [50]; (ii) a lack of qualified clinics or limited experience in Germany [31,51,52]; (iii) a lack of qualified information given to PwP [53,54]; (iv) patient reservations with fears [53,54], indecisiveness and the need for more time to decide (empirically about 1 year) [1,12] often resulting in DAT refusal [1,32]; or (v) contraindications or presentation of inappropriate candidates for DAT [32,55]. Furthermore, identification of candidates, management and handling of DAT, that finally relate to the treatment success [56], are often restricted to specialists. 

To overcome this, prospectively, screening tools such as “STIMULUS” (DBS [57]) or “MANAGE PD” (all DAT) might be helpful in identification and preselection of aPD with the need of DAT [4,32]. Moreover, comprehensive, patient-centered information material may support patients in the process of decision making and physicians in eloquent patient education [45]. Finally, educational material or trainings on aPD-relevant symptoms, indications and management of DAT for medical and nursing staff, such as “NAVIGATE PD” [50] or “Online Pflegeschule Parkinson” [58]) might also be advantageous, as well as the consultation of internationally already widely used specialist nurses (“PD nurses”) experienced in the identification and management of aPD including handling of DAT [56]. Overall, multidisciplinary approaches are desirable in this process to warrant optimal care. 

#### 4.2.2. Most Bothersome Symptoms of PwP with and without Suspected aPD

As expected, the aPD-suspect group was older, had an earlier disease onset, higher disease severity, lower ADL functioning, greater need for professional LTC and lower QoL, demonstrating their advanced disease stages and greater need for help compared to those not fulfilling aPD criteria. 

Interestingly, although akinesia/rigidity and autonomic dysfunctions were reported to be most bothersome in both PwP groups with and without suspected aPD, symptom profiles of both groups differed. The aPD group reported more often about motor fluctuations and falls, whereas the n-aPD group complained more about troublesome tremor. Thus, there might be some potential for the use of DAT in both groups as DAT show positive effects on the reported motor symptoms (akinesia/rigidity, motor fluctuations) as well as (refractory) tremor [34]. Intriguingly, therapy-resistant tremor is not included in the Delphi criteria (and also not considered in DAT preselection tools such as MANAGE PD), but can be sufficiently treated by DATs, especially with DBS [59]. Thus, aPD screening tools, information material and educational trainings on DATs should be extended to the use of DBS for therapy-resistant tremor. Furthermore, medical and nursing staff should be trained more intensively in identification and management of these different symptom constellations in different disease stages.

Altogether, proper symptom management including DAT application might result in better QoL and might possibly even reduce the need for LTC (or even prevent it) in aPD patients [14]. However, the management of axial symptoms such as falls—which are reported more often by the aPD group and are even predictors for the need of LTC [13]—is very challenging with limited effects of medication or DAT. Thus, alternative, non-pharmacological strategies may be needed here [60]; this is an aspect that is also true for many autonomic symptoms that have been described frequently by both PwP groups [61,62]. However, some autonomic symptoms may even respond to DATs with distinct profiles of different DAT types (DBS vs. infusion therapies) [36,63]. This highlights the parallel need for a greater focus on identification and management of non-motor (and axial) symptoms in PD patients in all disease stages and more individualized treatment strategies.

A limitation of the study is the relatively low response rate to the questionnaire that may bias the representativity of the study results. However, the response rate of 5.7% of this study is within the range of previous, comparable questionnaire studies of PD clientele [53,64,65], and resembled that of a recent questionnaire-based study (4.7%) using the same distribution method via the members’ journal of the German Parkinson Association but addressing a different topic [66]. Another limitation might be that the questionnaire was sent out to a preselected patient sample aiming for recruitment (mainly members of the German Parkinson association). Thus, results might not be generalized for the whole German PD patient community. However, the questionnaire was distributed nationwide, to PD patients across all sectors in rural and urban areas, with a wide range of age, age at diagnosis and disease severity, and included patients with and without professional care. This indicates the inclusion of a wide range of PD patients that might be representative for the German PD community to a certain degree.

Furthermore, results are based on subjective patient (or third person) information and can therefore be interpreted as an approximation or tendency instead of objective, “facts-based” data, but still, they give an important overview of the current situation. The interpretation of the Hoehn and Yahr scale, which is usually scored by medical staff, may be especially difficult regarding the lack of objective data and pure reliance on the individuals’ ability to assign their own disease severity.

Although we methodologically tried to create a barrier-free questionnaire that can easily be understood and avoids medical terms or too compartmentalized descriptions, proper comprehension and accuracy of patient answers cannot be guaranteed, especially in PwP with cognitive impairments. So, we cannot rule out a certain vagueness regarding some results. However, as third persons (e.g., family members, relatives, nursing staff or others) were allowed to support PwP in filling it out, we believe that answers might reflect reality although our design does not allow us to differentiate between patient and third person answers, which might result in a certain vagueness of some of our results.

## 5. Conclusions

Our data support that the DAT application rate in Germany is rather low with predominant use of DBS compared to infusion therapies. This contradicts international data and the fact that about every second PwP fulfills at least one of the aPD criteria, indicating a possible need of intensified treatment strategies. Furthermore, many symptoms perceived to be most bothersome by PwP with and without suspected aPD such as akinesia/rigidity, tremor, motor fluctuations and some autonomic dysfunctions could be tackled well with DAT. Thus, precise and early identification of these symptoms should be promoted and implemented in future DAT preselection tools and educational training programs for all disciplines involved in PD treatment and care. Furthermore, as some autonomic dysfunctions, but also axial symptoms such as falls, are rather therapy-resistant, future studies should focus on their proper identification and (non-pharmacological, non-DAT) management as well.

## Figures and Tables

**Figure 1 brainsci-13-00736-f001:**
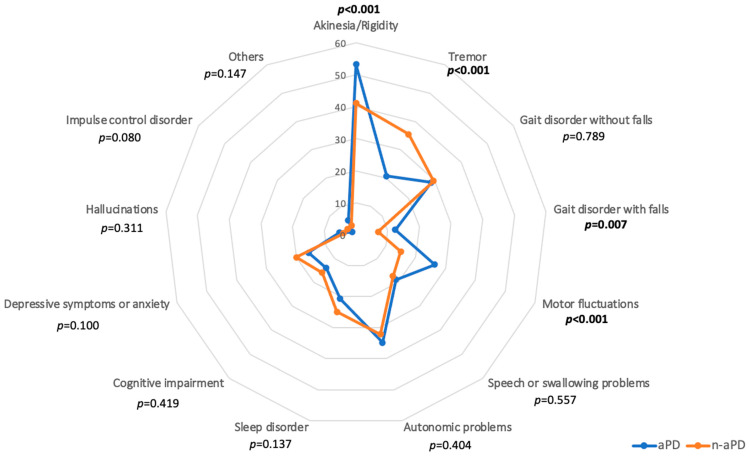
Parkinson’s symptoms rated as most bothersome (3 answers allowed, total *n* = 906) are depicted as percentage (%) of participants per group with (aPD, *n* = 492, blue) and without suspected advanced Parkinson’s disease (n-aPD, *n* = 414, orange). Right side: motor symptoms, left side: non-motor symptoms. Significant ^p^-values of group comparisons are given. “Others” includes, e.g., fatigue, pain, mental/concentration problems, vision problems, inner restlessness, freezing or camptocormia (free text answers). This figure refers to question #8 (see Appendix A). Data on dyskinesia (“involuntary movements”) are reported elsewhere in this manuscript.

**Table 1 brainsci-13-00736-t001:** Clinical and therapy-related data of people with Parkinson’s disease without DAT fulfilling aPD criteria (aPD group) or not (n-aPD group).

Patients without Device-Aided Therapies (*n* = 1116)					
Parameter	aPD (*n* = 627) (Mean [Min-Max] ± SD) or n (%)	Available Data	n-aPD (*n* = 489) (Mean [Min-Max] ± SD) or n (%)	Available Data	Statistics *p*-Value
Clinical Characteristics					
Gender (#3)	♂ 330 (53%) ♀ 290 (47%)	*n* = 620	♂ 262 (54%) ♀ 221 (46%)	*n* = 483	*p* = 0.736 ^b^
Age (years, #4)	73.6 (45–95) ± SD 8.3	*n* = 621	72.6 (43–91) ± SD 8.8	*n* = 481	***p* = 0.036 ^a^**
Age at diagnosis (years, #5)	62.9 (26–85) ± SD 9.6	*n* = 619	65.0 (22–87) ± SD 9.5	*n* = 477	***p* < 0.001 ^a^**
Hoehn and Yahr stage * (stages 1−5, #7)	3.3 (1–5) ± SD 1.1	*n* = 583	2.5 (1–5) ± SD 1.1	*n* = 457	***p* < 0.001 ^a^**
ADL (Katz index, total score 0−6, #13)	4.2 (0–6) ± SD 1.9	*n* = 556	5.3 (0–6) ± SD 1.2	*n* = 457	***p* < 0.001 ^a^**
Professional long-term care (LTC, #23)	LTC: 173 (27.6%) No LTC: 454 (72.4%)	*n* = 627	LTC: 62 (12.7%) No LTC: 427 (87.3%)	*n* = 489	***p* < 0.001 ^b^**
QoL (PDQ-8 total score, #10)	39.5 (0–100) ± SD 17.3	*n* = 515	25.6 (0–87.5) ± SD 16.8	*n* = 409	***p* < 0.001 ^a^**

Annotations: aPD = patients with suspected advanced Parkinson’s disease (fulfilling at least one of the aPD criteria). n-aPD = patients without suspected aPD. *n* = absolute number of patients. SD = standard deviation. ADL = activities of daily living according to the Katz index (total score of all six items). QoL = quality of life measured with the 8-item version of the Parkinson’s Disease questionnaire (PDQ-8; total score of all eight items). LTC = long-term care. Statistics: ^a^ Student *t*-test. ^b^ Pearson’s Chi squared test. Bold: statistically significant *p*-values (<0.05). # = question number in Care4PD patient survey (see Appendix A). * The Hoehn and Yahr stage was subjectively estimated by participants.

## Data Availability

The authors declare that all data supporting the findings of this study are available within the paper and its Appendix A.

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
