# Peer review of "Device-Aided Therapies in Parkinson’s Disease—Results from the German Care4PD Study"

_brainsci, 2023, doi:10.3390/brainsci13050736_

Round 1

Reviewer 1 Report

Comments and Suggestions for Authors

The authors performed a large, nationwide cross-sectional survey of Parkinson’s disease (PD) to clarify the 1) frequency and type of device-aided therapies (DAT), 2) frequent symptoms indicative of advanced PD, and 3) compared the most bothersome symptoms between advanced PD patients and non-advanced PD patients in Germany. The authors concluded that 12 % of PD patients received DAT (mainly deep brain stimulation), akinesia/rigidity and autonomic problems were most bothersome with and without suspected advanced PD patients. Non-advanced PD patients were more likely to have tremors, whereas advanced-PD patients were more likely to have fluctuations and fall.

 Although the present result might be significant, it should be cautious in interpreting present results.

The followings are my comments to the authors.

1.         As the authors suggested, the major limitation of this study is that results are based on subjective patient information. Please clarify how the authors instructed the patient about the symptoms of Parkinson's disease and how to validate the patient’s comprehension. For example, it might be difficult for the patients to identify the presence of stiffness. The gait and balance problems with and without falls are usually attributable to many pathological conditions and PD is not the only cause. The correct understanding of motor fluctuations is generally difficult for PD patients.

2.         The questionnaire regarding the most bothersome symptoms was originally developed. Please clarify how did authors validate the effectiveness of the originally developed questionnaire.

3.         The questionnaire concerning autonomic dysfunctions seems inappropriate. The nighttime urinary frequency and urinary urgency might be more frequent than incontinence. Furthermore, urinary incontinence is classified into urgent urinary incontinence, stress urinary incontinence, and overflow incontinence whose pathophysiology is quite different. The low blood pressure itself is not necessarily abnormal. Orthostatic hypotension rather than low blood pressure might be more appropriate.  

4.         The authors should clarify who answered the questionnaire. Although the motor and autonomic symptoms should be answered by the patient, memory problems and other similar problems should be answered by a caregiver.

Reviewer 2 Report

Comments and Suggestions for Authors

The study is very interesting and has its scientific and clinical merit, however some information is not very clear.

In the methods section, the second paragraph mentions the use of 56 questions, but at the end of this paragraph there is reference to a questionnaire attached in the supplementary material with only 13 questions. Which of the questionnaires were used?

In the research design, was any type of questioning foreseen or included to ensure understanding of what the patient was answering in the questionnaire? Because many of these patients may have some cognitive impairment and I would like to know if this could not have interfered with the accuracy of these patients' answers. Was this issue foreseen in the study design? If so, it is not mentioned in the methodological description.

As the questionnaire is the methodological focus of this study, each part of the questionnaire could be better detailed, the aspects analyzed so that the results are clearer. The authors detailed very well how the questionnaires were analyzed, but the content of the questionnaire was not clear.

The results presented in item 3.2.2. are interesting, but it is not clear how this information was collected, as it was not mentioned in the methodology how each item presented in the spinder chart was questioned. And some data presented in the 13 questions attached in the supplementary material was also not presented in the results section. I think that the description of the method and result should be more aligned, so that we know the methodology applied and how the result of the tool was.

The discussion and conclusion are coherent and adequate

Round 2

Reviewer 1 Report

Comments and Suggestions for Authors

I think that this manuscript has been revised correctly according to my suggestions.
